# Multi-Organ Protective Effects of Sodium Glucose Cotransporter 2 Inhibitors

**DOI:** 10.3390/ijms22094416

**Published:** 2021-04-23

**Authors:** Hidekatsu Yanai, Mariko Hakoshima, Hiroki Adachi, Hisayuki Katsuyama

**Affiliations:** Department of Diabetes, Endocrinology and Metabolism, National Center for Global Health and Medicine Kohnodai Hospital, 1-7-1 Kohnodai, Chiba 272-8516, Japan; d-hakoshima@hospk.ncgm.go.jp (M.H.); dadachidm@hospk.ncgm.go.jp (H.A.); d-katsuyama@hospk.ncgm.go.jp (H.K.)

**Keywords:** cardiovascular diseases, diabetes, hepatic fibrosis, nephropathy, sodium glucose cotransporter 2

## Abstract

Sodium glucose cotransporter 2 inhibitors (SGLT2i) block the reabsorption of glucose by inhibiting SGLT2, thus improving glucose control by promoting the renal excretion of glucose, without requiring insulin secretion. This pharmacological property of SGLT2i reduces body weight and improves insulin resistance in diabetic patients. Such beneficial metabolic changes caused by SGLT2i are expected to be useful not only for glucose metabolism, but also for the protection for various organs. Recent randomized controlled trials (RCTs) on cardiovascular diseases (EMPA-REG OUTCOME trial and CANVAS program) showed that SGLT2i prevented cardiovascular death and the development of heart failure. RCTs on renal events (EMPA-REG OUTCOME trial, CANVAS program, and CREDENCE trial) showed that SGLT2i suppressed the progression of kidney disease. Furthermore, SGLT2i effectively lowered the liver fat content, and our study demonstrated that SGLT2i reduced the degree of hepatic fibrosis in patients at high-risk of hepatic fibrosis. Such promising properties of SGLT2i for cardiovascular, renal, and hepatic protection provide us the chance to think about the underlying mechanisms for SGLT2i-induced improvement of multiple organs. SGLT2i have various mechanisms for organ protection beyond glucose-lowering effects, such as an increase in fatty acids utilization for hepatic protection, osmotic diuresis for cardiac protection, an improvement of insulin resistance for anti-atherogenesis, and an improvement of tubuloglomerular feedback for renal protection.

## 1. Introduction

Sodium glucose cotransporter 2 (SGLT2) is expressed in the proximal tubule of kidney and mediates the reabsorption of sodium and glucose [1]; therefore, SGLT2 inhibitors (SGLT2i) block the reabsorption of glucose by inhibiting SGLT2, thus improving glucose control by promoting the renal excretion of glucose, without requiring insulin secretion [2]. This pharmacological property of SGLT2i reduces body weight and improves insulin resistance in diabetic patients. Furthermore, these metabolic changes caused by SGLT2i are expected to be useful not only for glucose metabolism, but also for the protection of various organs.

In fact, recent randomized controlled trials (RCTs) present promising properties of SGLT2i for cardiovascular, renal, and hepatic protection. Here, we show accumulated studies indicating multi-organ protection by SGLT2i, and propose the underlying mechanisms for SGLT2i-induced improvement of multiple organs.

## 2. Anti-Atherogenic Properties of SGLT2i

We previously proposed possible anti-atherosclerotic effects beyond the glucose lowering of SGLT2i [3]. Briefly, caloric loss by SGLT2 inhibition may decrease plasma glucose without increasing insulin secretion, which may reduce body weight and result in an improvement of insulin resistance. An improvement of insulin resistance may decrease the blood triglyceride, inflammatory cytokines, and blood pressure (BP), and may increase high-density lipoprotein-cholesterol (HDL-C) and adiponectin [3]. Increased sodium excretion into urine and osmotic diuresis by SGLT2 inhibition may also decrease BP [3]. Various metabolic changes induced by SGLT2i may contribute to anti-atherosclerosis.

We previously studied patients with type 2 diabetes who had been continuously prescribed SGLT2i for 3 months or more, and compared the metabolic parameters before SGLT2i treatment with the data from 3 and 6 months after the SGLT2i treatment had started [4,5]. The SGLT2i treatment significantly reduced the body weight [4,5], systolic and diastolic BP [5], plasma glucose [5], hemoglobin A1c [4,5], and non-HDL-C [5], suggesting that SGLT2i improves atherosclerotic risk factors in patients with type 2 diabetes.

A systematic review and meta-analysis on the efficacy and safety of SGLT2i in patients with type 2 diabetes also showed that SGLT2i had a favorable effect on the hemoglobin A1c level (mean difference −0.66%; 95%CI (confidence interval) −0.73% to −0.58%) [6]. In this study, SGLT2i reduced the body weight (−1.80 kg; 95%CI −3.50 to −0.11 kg) and systolic BP (−4.45 mm Hg; 95%CI −5.73 to −3.18 mm Hg) [6].

Another systematic review and meta-analysis of 43 RCTs showed a significant reduction in systolic BP (−2.46 mm Hg; 95%CI −2.86 to −2.06), diastolic BP (−1.46 mm Hg; 95%CI −1.82 to −1.09), serum triglyceride (−2.08 mg/dL; 95%CI −2.51 to −1.64), and body weight (−1.88 kg; 95%CI −2.11 to −1.66) following treatment with SGLT2i [7].

A systematic review and meta-analysis of RCTs on the effect of SGLT2i on the blood adiponectin level in patients with type 2 diabetes showed that treatment with SGLT2i contributed to increased circulating adiponectin levels (+0.30; 95%CI 0.22 to 0.38) [8]. The effect of SGLT2i on inflammatory cytokines has not been reported at present. Adiponectin reduces inflammatory cytokines and oxidative stress [9]; therefore, an increase of adiponectin by SGLT2i may ameliorate a chronic inflammatory state in type 2 diabetes. The link between adipokines plasma levels and ischemic heart disease (IHD) in normal glucose tolerance (NGT) patients undergoing percutaneous coronary intervention (PCI) was assessed [10]. The adiponectin levels were independently associated to restenosis, de novo IHD, and overall new PCI, suggesting a protective role of adiponectin in coronary heart disease [10].

The EMPA-REG OUTCOME trial showed that empagliflozin reduced worsening nephropathy and progression to macroalbuminuria when compared with the placebo [11]. Canagliflozin also reduced the progression of albuminuria and suppressed the decline of the glomerular filtration rate (eGFR) when compared with the placebo in the CANVAS program [12]. Both the EMPA-REG OUTCOME trial and CANVAS program showed that SGLT2i significantly reduced death from cardiovascular causes and hospitalization for heart failure [12,13]. The presence of moderate proteinuria and diabetes is associated with a higher risk of mortality and cardiovascular events [14]. Albuminuria and chronic kidney disease can be the cause for secondary dyslipidemia [15]. Therefore, a reduction in the progression of albuminuria and a suppression of the decline of eGFR by SGLT2i may play a cardiovascular protective role.

The hypoglycemic effect of SGLT2i without increasing insulin secretion may result in body weight loss, which improves inflammatory state and insulin resistance, and induces an amelioration of oxidative stress, glucose and lipid metabolism, and BP (Figure 1). Furthermore, increased sodium excretion into urine and osmotic diuresis by SGLT2 inhibition may reduce BP and albuminuria and suppress eGFR decline. SGLT2i may play an anti-arteriosclerotic role through a such series of metabolic improvements.

## 3. Characteristic Glucose-Lowering Properties of SGLT2i

As the hypoglycemic effects of most oral anti-diabetic drugs depend on patients’ intrinsic insulin secretion and insulin resistance, the glucose-lowering effects of such drugs varies among patients. The most crucial determinant of the glucose-lowering effect of SGLT2i may be renal function. SGLT2i causes a decrease in the renal threshold for glucose excretion (RTG), an increase in urinary glucose excretion, and a reduction in plasma glucose levels in diabetic patients [16]. Therefore, if the plasma glucose levels were above RTG, SGLT2i was effective at decreasing the plasma glucose. However, if the plasma glucose levels were below RTG, SGLT2i did not work to decrease the plasma glucose. In type 1 diabetic patients treated with insulin and SGLT2i, glycemic control may be determined by an administered insulin dose and RTG reduced because of SGLT2i [17]. In type 2 diabetic patients treated with insulin and SGLT2i, glycemic control may be determined by an improvement of insulin resistance due to body weight loss, an improvement of glucotoxicity by insulin and/or SGLT2i, administered insulin dose, and RTG reduced because of SGLT2i [17].

To understand the influence of the estimated glomerular filtration rate (eGFR) on the improvement in metabolic parameters by SGLT2i, we analyzed our previous study [5]. We divided the patients into a high eGFR group (mean ± standard deviation (SD) of eGFR, 117 ± 36 mL/min/1.73 m^2^; n = 22) and the low eGFR group (72 ± 14 mL/min/1.73 m^2^; n = 26) by the mean value [18]. Hemoglobin A1c was significantly decreased in the high eGFR group when compared with the low eGFR group at 1, 2, 3, and 6 months after SGLT2i treatment started. The changes in hemoglobin A1c at 2 months (r = −0.361, *p* = 0.059, by the Pearson’s correlation) and 3 months (r = −0.349, *p* = 0.063) after the start of SGLT2i treatment tended to be negatively correlated with the baseline eGFR value. This result supports the significant contribution of renal function to the glucose-lowering effect of SGLT2i.

Furthermore, we studied the correlation of the pharmacokinetic properties of SGLT2i with the improvement in hemoglobin A1c [19]. We studied the correlation between the improvement in hemoglobin A1c by six kinds of SGLT2i (canagliflozin, ipragliflozin, tofogliflozin, empagliflozin, dapagliflozin, and luseogliflozin) with various pharmacological parameters, including daily standard dose, maximum plasma concentration (MC), SGLT2 inhibitory concentration 50% value (IC50), and SGLT2 selectivity, as well as the area under the blood concentration time curve (AUC) of each SGLT2i. A high IC50 means a low potency; therefore, we expressed “potency” by dividing the highest IC50 (ipragliflozin; 7.4) by the IC50 of each SGLT2i. MC × potency was the most crucial determinant of hemoglobin A1c reduction by SGLT2i after 12 weeks (r = −0.931, *p* = 0.007), suggesting that the potency to inhibit SGLT2 and patients’ renal SGLT2 integrity was the most crucial determinant of hemoglobin A1c reduction by SGLT2i.

## 4. Cardiovascular Protective Effects of SGLT2i

In EMPA-REG OUTCOME, empagliflozin reduced the primary outcome (death from cardiovascular causes, nonfatal myocardial infarction, or nonfatal stroke) by 14% when compared with the placebo [13]. In the empagliflozin group, there were significantly lower rates of death from cardiovascular causes (38% relative risk reduction) and hospitalization because of heart failure (35% relative risk reduction) than in the placebo. In the CANVAS program, the rate of the primary outcome (death from cardiovascular causes, nonfatal myocardial infarction, or nonfatal stroke) was lower by 14% with canagliflozin when compared with the placebo [12]. In the canagliflozin group there was a significantly lower rate of hospitalization for heart failure (33% relative risk reduction) than for the placebo. Both the EMPA-REG OUTCOME trial and CANVAS program showed that SGLT2i significantly reduced hospitalization for heart failure. However, almost all participants in the EMPA-REG OUTCOME trial had established cardiovascular diseases (CVD), and 66% of participants in the CANVAS program had a history of CVD. It remained unknown whether SGLT2i may reduce hospitalization for heart failure in type 2 diabetic patients without established CVD. The Comparative Effectiveness of Cardiovascular Outcomes in New Users of Sodium Glucose Cotransporter-2 Inhibitors (CVD-REAL) study, the first large real-world study of 309,056 type 2 diabetic patients, included patients with (13%) and without established CVD (87%), and included patients treated with canagliflozin (53%), dapagliflozin (42%), and empagliflozin (5%) [20]. In the CVD-REAL study, the SGLT2i treatment was significantly associated with a risk reduction (39%) in hospitalization for heart failure. In DECLARE-TIMI 58, 17,160 patients, including 10,186 without CVD, were followed for a median of 4.2 years [21]. Dapagliflozin did not result in a lower rate of major adverse cardiovascular events (MACE), but did result in a lower rate of cardiovascular death or hospitalization for heart failure (hazard ratio 0.83; 95%CI 0.73 to 0.95; *p* = 0.005), which reflected a lower rate of hospitalization due to heart failure (hazard ratio 0.73; 95%CI 0.61 to 0.88). Three RCTs and the large real-world study showed a significant contribution of SGLT2i to the prevention of heart failure [12,13,20,21].

In the EMPA-REG OUTCOME trial, there was a difference in hospitalization for heart failure between empagliflozin and placebo immediately after the start of the trial [13], leading us to consider the acute and chronic cardioprotective effects of SGLT2i [22]. Furthermore, the acute and chronic cardioprotective effects of SGLT2i may depend on hemodynamic and metabolic effects, respectively. A non-significant decrease in eGFR from the start of SGLT2i to 1 month after treatment, as well as a subsequent gradual increase in eGFR, were also observed in the EMPA-REG OUTCOME trial and in our previous study [11,23]. The phase showing a decline of eGFR may indicate acute effects of SGLT2i. Osmotic diuresis and increased urinary sodium excretion may induce body fluid depletion, which may result in reducing blood pressure and preventing the development of heart failure by acting like diuretics [22].

To understand which factor plays a crucial role in the prevention of heart failure in the early phase after the start of treatment with SGLT2i, we studied changes in plasma B-type natriuretic peptide (BNP), which is the marker for heart failure and other parameters in patients with type 2 diabetes [24]. According to the analysis of changes 3 months after the start of SGLT2i, the increase in the hematocrit levels in the BNP-increased group was significantly smaller than the BNP-decreased group. The change in plasma BNP levels was negatively and significantly correlated with the change in hematocrit levels. The larger increase in the hematocrit levels for the BNP-decreased group and a negative correlation between the changes in the plasma BNP and hematocrit levels 3 months after the start of SGLT2i indicate that osmotic diuresis may play an important role in the prevention of heart failure in the early phase after the start of SGLT2i [24]. A further analysis showed that eGFR decreased in the BNP-decreased group, and eGFR increased in the BNP-increased group. The change in plasma BNP levels was positively and significantly correlated with the change in eGFR. The serum blood urea nitrogen (BUN) increased in the BNP-decreased group, and BUN decreased in the BNP-increased group. The change in plasma BNP levels tended to be negatively correlated with the change in plasma BUN levels. These data also indicate a significant contribution of SGLT2i-induced body fluid depletion for prevention of heart failure in the early phase.

Chronic effects of SGLT2i for the prevention of heart failure include an improvement in myocardial energetics, reduction of albuminuria, suppression of eGFR decline, reduced sympathetic overactivity, and anti-atherosclerotic effects [22,25,26]. SGLT2i may induce relative glucose deficiency, and may then trigger increased lipolysis and fatty acids (FA) oxidation, which increase hepatic ketone bodies’ production [27]. Under conditions of persistent mild hyperketonemia during treatment with SGLT2i, β-hydroxybutyrate is freely taken up by the heart and is preferentially oxidized over FA and glucose [28]. SGLT2i may improve the efficiency of myocardial energetics by offering β-hydroxybutyrate as an attractive fuel for oxidation [29]. The cardiorenal benefits of SGLT2i may be due to a shift in myocardial and renal fuel metabolism away from fat and glucose oxidation, which are energy inefficient in the diabetic heart and kidney, toward an energy-efficient super fuel like ketone bodies [30]; however, this finding is controversial and has not been consistently replicated in all studies. We also need to consider the adverse events of increased ketone body production due to SGLT2i. We should pay an attention to the development of diabetic ketoacidosis when we use SGLT2i for patients who were diagnosed with type 2 diabetes and who were subsequently found to have latent autoimmune diabetes of adulthood, patients who had recently undergone major surgery, or patients who had decreased or discontinued insulin [31]. Both the EMPA-REG OUTCOME trial and CANVAS program showed that SGLT2i reduced albuminuria and suppressed eGFR decline when compared with the placebo [11,12]. A reduction in the progression of albuminuria and the suppression of a decline of eGFR by SGLT2i may contribute to prevention of heart failure. Chronic activation of the sympathetic nervous system has been identified in heart failure [32], and sympathetic overactivity is associated with a poor prognosis in patients with heart failure [33]. SGLT2i has been reported to reduce sympathetic overactivity [26], contributing to the prevention of heart failure.

Cowie, MR. et al. listed early natriuresis with a reduction in plasma volume, improved vascular function, a reduction in BP and changes in tissue sodium handling, a reduction in adipose tissue-mediated inflammation and pro-inflammatory cytokine production, a shift towards ketone bodies as the metabolic substrate for the heart and kidneys, reduced oxidative stress, lowered serum uric acid level, reduced glomerular hyperfiltration and albuminuria, and suppressed advanced glycation end-product signaling as the beneficial mechanisms of SGLT2i for cardiovascular and renal outcomes [34].

Furthermore, Joshi, SS. et al. suggested improved glycaemic control, diuresis, weight reduction and a reduction in blood pressure, improved cardiomyocyte calcium handling, enhanced myocardial energetics, induced autophagy, and reduced epicardial fat as the proposed mechanisms of SGLT2i action in cardiovascular health and disease [35].

## 5. Renal Protective Effects of SGLT2i

In the EMPA-REG OUTCOME trial, the incident or worsening nephropathy, doubling of the serum creatinine level, initiation of renal-replacement therapy, and progression to macroalbuminuria reduced by 39%, 44%, 55%, and 38% in the empagliflozin group when compared with the placebo group, respectively [11]. A possible benefit of canagliflozin with respect to the progression of albuminuria (hazard ratio 0.73; 95%CI 0.67 to 0.79) and the composite outcome of a sustained 40% reduction in the eGFR, the need for renal-replacement therapy, or death from renal causes (hazard ratio 0.60; 95%CI 0.47 to 0.77) were observed in the CANVAS program [12]. The CREDENCE trial included patients with type 2 diabetes and albuminuric chronic kidney disease who received either canagliflozin or a placebo (n = 4401) [36]. After a follow-up of 2.62 years, the relative risk for the renal-specific composite of end-stage kidney disease, a doubling of the creatinine level, or death from renal causes was lower by 34%, and the relative risk of end-stage kidney disease was lower by 32% in the canagliflozin group than in the placebo group.

Possible renal protective mechanisms by SGLT2i are shown in Figure 2 [23,37,38]. An improvement in metabolic factors, such as reduced body weight, insulin resistance, and serum uric acid, may also contribute to renal protection by SGLT2i. An increase in the utilization of ketone bodies by diabetic failing renal cells and an improvement in cardiac function may be beneficially associated with renal protection.

Osmotic diuresis by SGLT2i decreases blood pressure, which is beneficial for renal protection. Sano et al. mentioned that SGLT2i reduces the overload of the proximal tubules and improves tubulointerstitial hypoxia, inducing the recovery of erythropoietin production by fibroblasts [39]. They concluded that increased hematocrit during SGLT2i therapy indicates the recovery of tubulointerstitial function in diabetic kidneys [39]. We think that elevated erythropoietin, which increases hematocrit, may be also a possible mechanism for the renal protective effect of SGLT2i [23]. Chronic treatment with recombinant human erythropoietin exerted renal protective effects beyond hematopoiesis in streptozotocin-induced diabetic rats [40]. Erythropoietin protected mouse podocytes from damage by advanced glycation end-products [41]. Erythropoietin ameliorated podocyte injury in advanced diabetic nephropathy in db/db mice [42]. In humans, serum erythropoietin transiently increased from baseline in the dapagliflozin group, up until week 4 [43]. These animal and human studies support our hypothesis. Heerspink H.J.L. described the sodium-related physiological effects of SGLT2i and the impact on kidney protection [44]. An increased SGLT2 mRNA expression increased the renal NaCl reabsorption in the proximal tubule, leading to a marked reduction in distal NaCl delivery to the macula densa [45]. According to the tubular hypothesis for glomerular hyperfiltration, the decline in macula densa NaCl delivery was sensed as a reduction in the circulating plasma volume by the juxtaglomerular apparatus, which is called tubuloglomerular feedback, and led to maladaptive glomerular afferent arterial vasodilatation and increased intraglomerular pressure [46]. SGLT2 inhibition increased distal renal NaCl delivery, causing an increased afferent tone, thereby reducing the intraglomerular pressure and glomerular hyperfiltration [47], which may reduce albuminuria and suppress the decline of eGFR.

## 6. Hepatic Protective Effects of SGLT2i

We previously reported that SGLT2i improved liver function, in addition to lowering plasma glucose [4,5]. When we consider the mechanisms for hepatic protection by SGLT2i, we should think about the effects of SGLT2i on FA metabolism in the adipose tissue, skeletal muscle, and liver [48].

Empagliflozin effectively lowered the liver fat content in type 2 diabetic patients [49]. In this study, empagliflozin raised adiponectin levels [49], which has beneficial effects on glucose and lipid metabolism through the activation of adenosine 5’-monophosphate (AMP)-activated protein kinase (AMPK) [9]. Furthermore, AMPK activation was induced by canagliflozin, which was caused by the inhibition of Complex I of the respiratory chain [50]. Acetyl-CoA carboxylase (ACC) can be regulated at the level of gene expression, allosteric regulation of the enzyme, and reversible phosphorylation by AMPK [51]. Its inactivation in the heart and skeletal muscle through phosphorylation by AMPK is becoming well-established. ACC is an important target of certain hypolipidemic drugs, such as the fibrates. This is not simply because ACC alpha inhibition decreases the synthesis of FA in liver; it is also because ACC beta inhibition leads to a decrease in malonyl-CoA levels and to a disinhibition of FA oxidation [51]. Canagliflozin inhibited lipid synthesis, an effect that was absent in AMPK knockout cells and that required the phosphorylation of ACC at the AMPK sites [50]. SGLT2i ameliorated fat deposition and increased AMPK phosphorylation, resulting in phosphorylation of its major downstream target, ACC, which led to the downregulation of downstream FA synthesis-related molecules and to the upregulation of downstream β oxidation-associated molecules [52]. Tofogliflozin reduced the body weight gain, mainly because of fat mass reduction associated with a diminished adipocyte size [53]. Serum free FA and ketone bodies were increased in the tofogliflozin-treated mice, suggesting the acceleration of lipolysis in adipose tissues and hepatic β-oxidation [53]. Tofogliflozin ameliorated insulin resistance by increasing the glucose uptake in the skeletal muscle. Empagliflozin shifted energy metabolism towards FA utilization, elevated AMPK, and ACC phosphorylation in the skeletal muscle in diet-induced obese mice [54]. SGLT2i induced a negative energy balance state by excreting glucose in the urine, which may induce alterations in the glucose–FA cycle [55]. The fundamental concept of the glucose–FA cycle is reciprocal substrate competition between glucose and FA in oxidative tissues such as the skeletal muscle. We speculate that SGLT2i-mediated the alteration of the glucose–FA cycle may induce changes in glucose and FA metabolism in the skeletal muscle, adipose tissue, and liver, which may be associated with the amelioration of liver function.

We previously attempted to elucidate the effects of SGLT2i on hepatic fibrosis in patients with type 2 diabetes [56]. Hepatic fibrosis was evaluated using the noninvasive fibrosis-4 (FIB4) index, which was already reported as a useful index in nonalcoholic fatty liver diseases [57,58]. We enrolled 315 patients in this study. In the analysis of all of the cases, a significant change in the FIB4 index was not observed. We divided the studied patients into three groups according to the baseline FIB4 index. Only in the group with a high value for the baseline FIB4 index, the FIB4 index was significantly decreased after 12 months. The correlations between the change of FIB4 index during the 12-month SGLT2i treatment was inversely correlated with the baseline FIB4 index. Our study demonstrated that SGLT2i ameliorated fibrosis in the liver in patients at high risk of hepatic fibrosis.

The summary of possible hepatic protective mechanisms by SGLT2i is shown in Figure 3. Suppression of the renal glucose reabsorption reduce body weight and insulin resistance, and increased adiponectin, which induced AMPK activation. AMPK activation increased glucose uptake by the skeletal muscle and inactivated ACC, which decreased FA synthesis and increased FA oxidation in the liver. Increased renal excretion of glucose may alter the glucose–FA cycle and may result in an increase in FA use/oxidation in the skeletal muscle and liver. Insulin resistance increases the activity and expression of hormone-sensitive lipase in adipose tissue, which catalyzes the breakdown of triglyceride (lipolysis), releasing FA [59], and an improvement of insulin resistance reduces FA release. As SGLT2i improves insulin resistance, enhanced lipolysis by SGLT2i cannot be explained by insulin resistance. Enhanced lipolysis may be induced by the promotion of the use of FA, instead of glucose lost by SGLT2i as an energy source. Such altered FA metabolism in the adipose tissue, skeletal muscle, and liver by SGLT2i may reduce hepatic FA accumulation, resulting in a reduction of inflammation and oxidative stress, which may contribute to an improvement of liver function.

## 7. Conclusions

Unique multi-organ protective effects beyond the hypoglycemic effect of SGLT2i are shown in Figure 4. SGLT2i has various mechanisms for organ protection beyond glucose-lowering effects, such as an increase in FA utilization for hepatic protection, osmotic diuresis for cardiac protection, an improvement in insulin resistance for anti-atherogenesis, and an improvement of tubuloglomerular feedback for renal protection.

## Figures and Tables

**Figure 1 ijms-22-04416-f001:**
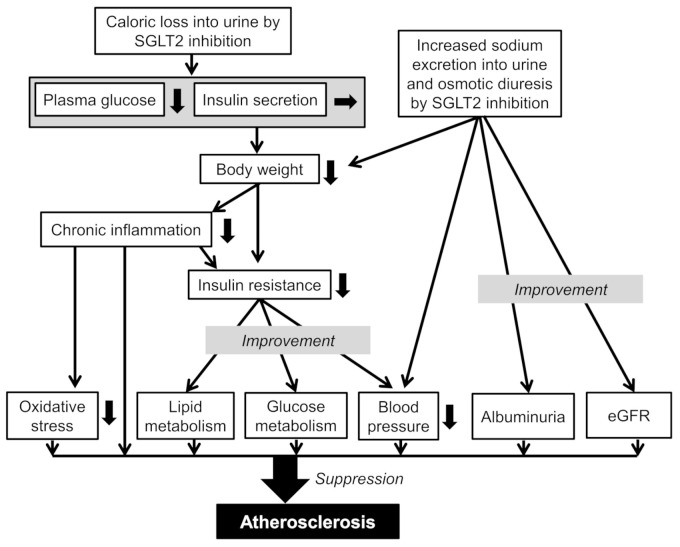
Possible anti-atherosclerotic mechanisms of sodium glucose cotransporter 2 inhibitors (SGLT2i) inhibitors. The bold down arrows indicate a decrease.

**Figure 2 ijms-22-04416-f002:**
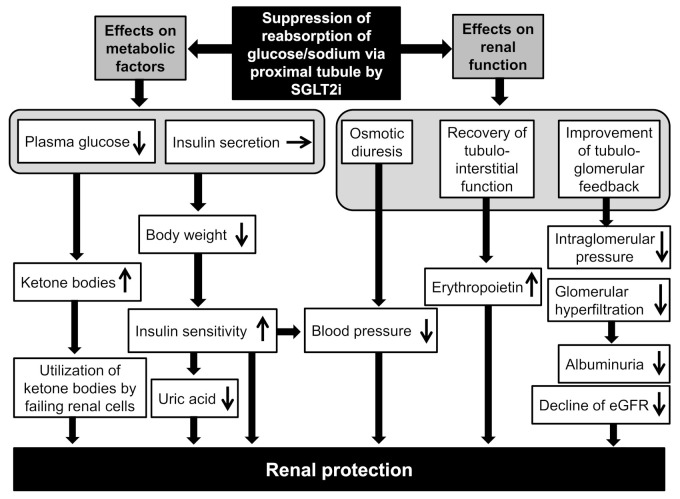
Possible renal protective mechanisms of SGLT2 inhibitors. The up and down arrows in white boxes indicate an increase and a decrease, respectively.

**Figure 3 ijms-22-04416-f003:**
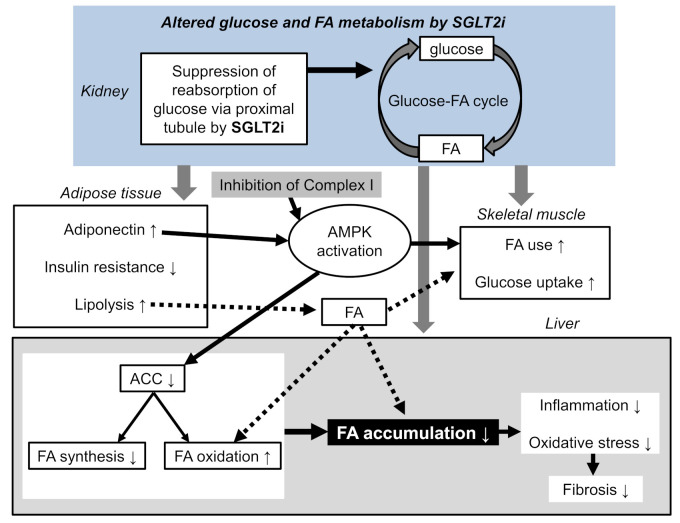
Possible hepatic protective mechanisms by SGLT2 inhibitors. ACC, acetyl-CoA carboxylase; AMPK, adenosine monophosphate-activated protein kinase; FA, fatty acids. The up and down arrows in white boxes indicate an increase and a decrease, respectively.

**Figure 4 ijms-22-04416-f004:**
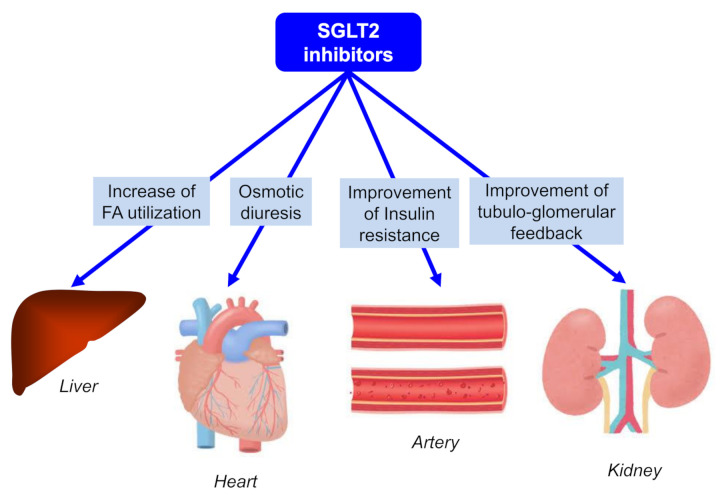
Unique multi-organ protective effects beyond the hypoglycemic effect of SGLT2 inhibitors.

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
