# Peer review of "Multi-Organ Protective Effects of Sodium Glucose Cotransporter 2 Inhibitors"

_ijms, 2021, doi:10.3390/ijms22094416_

Round 1

Reviewer 1 Report

The authors are reviewing research in this area, primarily introducing their studies and also referring to larger research studies.

However, there are some concerns. There are some more comprehensive reviews in this area, for example
https://pubmed.ncbi.nlm.nih.gov/27931088/
https://www.nature.com/articles/s41569-020-0406-8
https://heart.bmj.com/content/early/2021/02/25/heartjnl-2020-318060
but the authors have not touched on these at all. This is an unfair attitude.
The authors claim that "the fuel selection improves the transduction of oxygen consumption into work efficiency at the mitochondrial level" by ketone bodies production, but the review of Bonora and Avogaro is more objective, introducing that "although this finding is controversial and has not been consistently replicated in all studies".
It's clear which is more objective, and which is beneficial to the reader.

When there is an existing review, the value of the new review will be when it provides new insights and ideas. Relating to this, there are two things to worry about.

1. The authors often refer to "our hypothesis". Such as, "Our previous study demonstrated that the addition of SGLT2i to intensive insulin therapy reduced prandial insulin doses and body weight in type 2 diabetic patients [12], supporting our hypothesis." and "These animal and human studies support our hypothesis."
However, it is not clear what kind of hypothesis it is. For example, in the third paragraph, which begins with ”We previously proposed”, no new clear claims were found. It seems that it was planning to write the new hypothesis somewhere, but haven't it forgotten, or is this my oversight?

2. In the summary, I read that "Furthermore, SGLT2i effectively lowered liver fat content, and our study demonstrated that SGLT2i
reduced the degree of hepatic fibrosis in high-risk patients for hepatic fibrosis." So I expected new findings and ideas about hepatic fibrosis, but they only introduced a study reporting that there seems to be a significant difference under certain conditions. Any detailed mechanisms seem to be still unknown.

Frankly, this was the only original thing I could find that no other reviews described. If this is true, how meaningful that to re-introduce it in the form of a review? 

The main reason I had to read this manuscript suspiciously was that this review completely ignored the preceding reviews. They have to be quoted.  The contents should be completely rewritten so that any new knowledge and new insights become clear.

Minor points

Figure 2. 
The undefined vertical lines in this figure are a decisive omission. Also, the change is so small that it seems to be meaningless as evidence. Isn't the authors' claim that this amount of change important is somewhat unreasonable?

Figure 4.
It seems strange that the mitochondria and the organs are written in the same line. Also, FA oxidation is in a position unrelated to the mitochondria is strange. Additionally, the explanation that FA oxidation rises when ACC is lowered makes readers worried that the authors may have misunderstood the function of ACC. Also, the way to connect the phenomena is too casual. If they have evidence, they should be introduced; otherwise, it's just an imagination. The figure should be reconstructed. 

Author Response

  1. According to the comment “However, there are some concerns. There are some more comprehensive reviews in this area, for example https://pubmed.ncbi.nlm.nih.gov/27931088/

https://www.nature.com/articles/s41569-020-0406-8

https://heart.bmj.com/content/early/2021/02/25/heartjnl-2020-318060

but the authors have not touched on these at all. This is an unfair attitude.

We added sentences by citing all articles as the followings.

We also need to consider the adverse events of increased ketone body production due to SGLT2i. We should pay an attention to the development of diabetic ketoacidosis, when we use SGLT2i for patients who were diagnosed with type 2 diabetes and were subsequently found to have latent autoimmune diabetes of adulthood, patients who had recently undergone major surgery, or patients who had decreased or discontinued insulin [31].

  1. Burke, KB.; Schumacher, CA.; Harpe, SE. 1SGLT2 Inhibitors: A Systematic Review of Diabetic Ketoacidosis and Related Risk Factors in the Primary Literature. Pharmacotherapy 2017, 37, 187-194.

Cowie, MR. et al. listed early natriuresis with a reduction in plasma volume, improved vascular function, a reduction in BP and changes in tissue sodium handling, a reduction in adipose tissue-mediated inflammation and pro-inflammatory cytokine production, a shift towards ketone bodies as the metabolic substrate for the heart and kidneys, reduced oxidative stress, lowered serum uric acid level, reduced glomerular hyperfiltration and albuminuria, and suppression of advanced glycation end-product signalling as the beneficial mechanisms of SGLT2i for cardiovascular and renal outcomes [34].

Further, Joshi, SS., et al. suggested improved glycaemic control, diuresis, weight reduction and reduction in blood pressure, improved cardiomyocyte calcium handling, enhanced myocardial energetics, induced autophagy and reduced epicardial fat as the proposed mechanisms of SGLT2i action in cardiovascular health and disease [35].

  1. Cowie, M.R.; Fisher, M. SGLT2 inhibitors: mechanisms of cardiovascular benefit beyond glycaemic control. Nature Reviews Cardiology 2020, 17, 761–772.

  1. Joshi, S.S.; Singh, T.; Newby, D.E.; Singh, J. Sodium-glucose co-transporter 2 inhibitor therapy: mechanisms of action in heart failure. 2021 Feb 26:heartjnl-2020-318060.

  1. According to the comment “The authors claim that "the fuel selection improves the transduction of oxygen consumption into work efficiency at the mitochondrial level" by ketone bodies production, but the review of Bonora and Avogaro is more objective, introducing that "although this finding is controversial and has not been consistently replicated in all studies".It's clear which is more objective, and which is beneficial to the reader.”

We changed to

SGLT2i may induce relative glucose deficiency, and then may trigger increased lipolysis and fatty acids (FA) oxidation which increase hepatic ketone bodies production [27]. Under conditions of persistent mild hyperketonemia during treatment with SGLT2i, β-hydroxybutyrate is freely taken up by the heart and oxidized in preference to FA and glucose [28]. SGLT2i may improve the efficiency of myocardial energetics by offering β-hydroxybutyrate as an attractive fuel for oxidation [29]. The cardiorenal benefits of SGLT2i may be due to a shift in myocardial and renal fuel metabolism away from fat and glucose oxidation, which are energy inefficient in diabetic heart and kidney, toward an energy-efficient super fuel like ketone bodies [30], however, this finding is controversial and has not been consistently replicated in all studies.

  1. According to the comment “The authors often refer to "our hypothesis". Such as, "Our previous study demonstrated that the addition of SGLT2i to intensive insulin therapy reduced prandial insulin doses and body weight in type 2 diabetic patients [12], supporting our hypothesis." and "These animal and human studies support our hypothesis."

 We deleted "Our previous study demonstrated that the addition of SGLT2i to intensive insulin therapy reduced prandial insulin doses and body weight in type 2 diabetic patients [12], supporting our hypothesis."

  1. According to the comment However, it is not clear what kind of hypothesis it is. For example, in the third paragraph, which begins with ”We previously proposed”, no new clear claims were found. It seems that it was planning to write the new hypothesis somewhere, but haven't it forgotten, or is this my oversight?

We added the following sentences.

Briefly, caloric loss by SGLT2 inhibition may decrease plasma glucose without increasing insulin secretion, which may reduce body weight and result in improvement of insulin resistance. An improvement of insulin resistance may decrease blood triglyceride, inflammatory cytokines and blood pressure (BP), and may increase high-density lipoprotein-cholesterol (HDL-C) and adiponectin [3]. Increased sodium excretion into urine and osmotic diuresis by SGLT2 inhibition may also decrease BP [3]. Various metabolic changes induced by SGLT2i may contribute to anti-atherosclerosis.

  1. According to the comment “In the summary, I read that "Furthermore, SGLT2i effectively lowered liver fat content, and our study demonstrated that SGLT2i reduced the degree of hepatic fibrosis in high-risk patients for hepatic fibrosis." So I expected new findings and ideas about hepatic fibrosis, but they only introduced a study reporting that there seems to be a significant difference under certain conditions. Any detailed mechanisms seem to be still unknown.”

We added the following sentences and re-made Figure 4.

The summary of possible hepatic protective mechanisms by SGLT2i were shown in Figure 4. Suppression of renal glucose reabsorption reduces body weight and insulin resistance, and increases adiponectin, which induce AMPK activation. AMPK activation increase glucose uptake by skeletal muscle and inactivates ACC which decreases FA synthesis and increases FA oxidation in liver. Increased renal excretion of glucose may alter glucose-FA cycle and may result in increase of FA use/oxidation in skeletal muscle and liver. Insulin resistance increases activity and expression of hormone-sensitive lipase in adipose tissue, which catalyzes the breakdown of triglyceride (lipolysis), releasing FA [59], and an improvement of insulin resistance reduces FA release. Since SGLT2i improve insulin resistance, enhanced lipolysis by SGLT2i cannot be explained by insulin resistance. Enhanced lipolysis may be induced by the promotion of use of FA instead of glucose lost by SGLT2i as an energy source. Such altered FA metabolism in adipose tissue, skeletal muscle and liver by SGLT2i may reduce hepatic FA accumulation, resulting in reduction of inflammation and oxidative stress, which may contribute to an improvement of liver function.

  1. According to the comment “Figure 2. The undefined vertical lines in this figure are a decisive omission. Also, the change is so small that it seems to be meaningless as evidence. Isn't the authors' claim that this amount of change important is somewhat unreasonable?”

We added the following sentences.

Figure 2 showed the change of eGFR after the start of SGLT2i which was observed in our study [23]. Such a rapid decrease in eGFR from the start of SGLT2i to 1 month and a subsequent gradual increase in eGFR have also been observed in EMPA-REG OUTCOME trial [11]. The phase showing a rapid decrease of eGFR may indicate acute effects of SGLT2i.

  1. According to the comment “Figure 4. It seems strange that the mitochondria and the organs are written in the same line. Also, FA oxidation is in a position unrelated to the mitochondria is strange. Additionally, the explanation that FA oxidation rises when ACC is lowered makes readers worried that the authors may have misunderstood the function of ACC. Also, the way to connect the phenomena is too casual. If they have evidence, they should be introduced; otherwise, it's just an imagination. The figure should be reconstructed.”

We added the following sentences and re-made Figure 4.

Acetyl-CoA carboxylase (ACC) can be regulated at the level of gene expression, allosteric regulation of the enzyme, and reversible phosphorylation by AMPK [51]. Its inactivation in heart and skeletal muscle through phosphorylation by AMPK is becoming well-established. ACC is an important target of certain hypolipidemic drugs such as the fibrates. This is not simply because ACC alpha inhibition decreases the synthesis of FA in liver; it is also because ACC beta inhibition leads to a decrease in malonyl-CoA levels and the disinhibition of FA oxidation [51].

Reviewer 2 Report

In this narrative review, Hidekatsu Yanai et al. described the protective effects of SGLT2i on several target, beyond glucose-lowering effects. In particular, the authors investigated the role of SGLT2i as organ protectors towards the heart, arterial circulation, kidney and liver.

I enjoyed reading this interesting manuscript. The topic is hot. The document is well written, the structure of the review is correct. The references are updated. The figures are clear.

However, this reviewer raises some criticisms that the authors have to address.

1- During the review, a brief reference to the effect of SGLT2i on proteinuria is made only at the beginning of section 5, where the authors comment on the effects on composite renal endpoints in some RCTs (EMPAREG, CANVAS, CREDENCE). Actually, the antiproteinuric effect of SGLT2i is not only extremely evident in all RCTs, but above all it is a mechanism that could justify the reduction of both cardiovascular and renal risk in type 2 diabetes. In fact, the impact of the modification of proteinuria on cardio-renal risk has been documented by recent real-life studies (Nephrol Dial Transplant. 2018 Nov 1; 33 (11): 1942-1949. Doi: 10.1093 / ndt / gfy032.). This important issue should be commented on in the text, both in paragraph 4 and paragraph 5, and this reference should be added. An integration in Figures 1 and 3 would also be useful.

2- The authors at page 2 write: “A systematic review and meta-analysis of RCTs on the effect of SGLT2i on blood adiponectin level in patients with type 2 diabetes showed that the treatment with SGLT2i contributed to an increased circulating adiponectin levels….. Adiponectin reduces inflammatory cytokines and oxidative stress, therefore, an increase of adiponectin by SGLT2i may ameliorate chronic inflammatory state in type 2 diabetes.” This statement is absolutely true. In particular, the protective role of adiponectin in coronary heart disease was recently well documented (Cardiovasc Diabetol. 2019 Mar 4; 18 (1): 24. doi: 10.1186 / s12933 - 019-0826-0). This issue and this reference should be commented in the manuscript.

3- The manuscript needs a linguistic revision by a native English speaker.

Author Response

  1. According to the comment “During the review, a brief reference to the effect of SGLT2i on proteinuria is made only at the beginning of section 5, where the authors comment on the effects on composite renal endpoints in some RCTs (EMPAREG, CANVAS, CREDENCE). Actually, the antiproteinuric effect of SGLT2i is not only extremely evident in all RCTs, but above all it is a mechanism that could justify the reduction of both cardiovascular and renal risk in type 2 diabetes. In fact, the impact of the modification of proteinuria on cardio-renal risk has been documented by recent real-life studies (Nephrol Dial Transplant. 2018 Nov 1; 33 (11): 1942-1949. Doi: 10.1093 / ndt / gfy032.). This important issue should be commented on in the text, both in paragraph 4 and paragraph 5, and this reference should be added. An integration in Figures 1 and 3 would also be useful.”

We added the following sentences by citing recent real-life studies (Nephrol Dial Transplant. 2018 Nov 1; 33 (11): 1942-1949. Doi: 10.1093 / ndt / gfy032.).

EMPA-REG OUTCOME trial showed that empagliflozin reduced worsening nephropathy and progression to macroalbuminuria as compared with placebo [11]. Canagliflozin also reduced progression of albuminuria and suppressed a decline of the glomerular filtration rate (eGFR) as compared with placebo in the CANVAS program [12]. Both EMPA-REG OUTCOME trial and CANVAS program showed that SGLT2i significantly reduced death from cardiovascular causes and hospitalization for heart failure [12,13]. The presence of moderate proteinuria and diabetes is associated with a higher risk of mortality and cardiovascular events [14]. Albuminuria and chronic kidney disease can be the cause for secondary dyslipidemia [15]. Therefore, reduction of progression of albuminuria and suppression of a decline of eGFR by SGLT2i may play a cardiovascular protective role.  

We re-made Figure 1, 2, 3 by including reduction of progression of albuminuria and suppression of a decline of eGFR.

  1. According to the comment “The authors at page 2 write: “A systematic review and meta-analysis of RCTs on the effect of SGLT2i on blood adiponectin level in patients with type 2 diabetes showed that the treatment with SGLT2i contributed to an increased circulating adiponectin levels….. Adiponectin reduces inflammatory cytokines and oxidative stress, therefore, an increase of adiponectin by SGLT2i may ameliorate chronic inflammatory state in type 2 diabetes.” This statement is absolutely true. In particular, the protective role of adiponectin in coronary heart disease was recently well documented (Cardiovasc Diabetol. 2019 Mar 4; 18 (1): 24. doi: 10.1186 / s12933 - 019-0826-0). This issue and this reference should be commented in the manuscript.”

We added the following sentences by citing (Cardiovasc Diabetol. 2019 Mar 4; 18 (1): 24. doi: 10.1186 / s12933 - 019-0826-0).

The link between adipokines plasma levels and ischemic heart disease (IHD) in Normal Glucose Tolerance (NGT) patients undergoing Percutaneous Coronary Intervention (PCI) was assed [10]. Adiponectin levels were independently associated to restenosis, de novo IHD and overall new PCI, suggesting a protective role of adiponectin in coronary heart disease [10].

Round 2

Reviewer 1 Report

The manuscript has become easier to understand and fairer than the original. I would like to pay tribute to the efforts of the authors.

However, one problem remains. The authors have not clarified what the whiskers in Fig. 2 are. Do they represent, for example, the standard deviation of measurements on each time period?

If the whiskers represent standard deviation or standard error, it seems that there is no significant difference in the increase or decrease in eGFR claimed by the authors; the range of increase/decrease is much smaller than the whiskers. If these whiskers represent something different, they should be revealed.

I would like to ask the authors to use a Box plot for a more polite presentation. And if the range of this increase or decrease is really this much, then more data is needed for this claim.

Author Response

Reviewer 1

  1. According to the comment “The authors have not clarified what the whiskers in Fig. 2 are. Do they represent, for example, the standard deviation of measurements on each time period? If the whiskers represent standard deviation or standard error, it seems that there is no significant difference in the increase or decrease in eGFR claimed by the authors; the range of increase/decrease is much smaller than the whiskers. If these whiskers represent something different, they should be revealed.”

As the reviewer pointed out, both our study and EMPA-REG OUTCOME trial did not show a statistically significant difference in the decrease in eGFR up to 1 month after SGLT2i administration. Therefore, we deleted Figure 2 and added the following sentence.

A non-significant decrease in eGFR from the start of SGLT2i to 1 month after and a subsequent gradual increase in eGFR have also been observed in EMPA-REG OUTCOME trial and our previous study [11,23].

Reviewer 2 Report

None

Author Response

None